# The impact of Covid-19 pandemic on overall well-being of practicing lawyers

**Michael Fore** [1]*, **Erin Stevenson**[2]

**1** College of Business, Eastern Kentucky University, Richmond, KY, United States of America, **2** Eastern Kentucky University, Richmond, KY, United States of America

* michael.fore@eku.edu

## Abstract

Every day lawyers provide counsel and advocacy to individuals, groups, and businesses in a multitude of settings. From court room to board room, attorneys are relied upon to guide their clients through difficult situations. In doing this, attorneys all too often internalize the stresses of those that they help. The legal system has long been considered a stressful occupation. This stressful environment was further taxed by the wider societal disruptions in 2020 as we dealt with the onset of the COVID-19 pandemic. Beyond the illness itself, the pandemic forced widespread court closures and made it more difficult to communicate with clients. Based upon a survey of the membership of the Kentucky Bar Association, this paper considers the impact of the pandemic on attorney wellness in a variety of categories. These results demonstrated marked negative impacts on a variety of wellness measures which may result in significant reductions in service provision and efficacy for the people who need legal services. The pandemic made the practice of law harder and more stressful. Attorneys suffered increased incidence of substance abuse, alcohol consumption, and stress during the pandemic. These results were generally worse among those practicing in the areas of criminal law. In light of these adverse psychological effects facing attorneys, the authors argue the need for increased mental health support resources for attorneys, as well as establishing clear steps to raise awareness among the legal community about the importance of mental health and personal wellness.

## Introduction

Lawyers play a unique role in society. They serve as counselors and advisors to individuals and businesses. Many people only deal with lawyers during the worst events of their life- a divorce, a criminal citation, or personal injury. Yet lawyers also counsel entrepreneurs on their new ventures, advise inventors on their creations, and steer individuals and businesses through their most important financial transactions. Legal services are a vital part of a functioning and equitable civil society as well as necessary for a vibrant economy. Throughout North America the legal system serves as a framework and set of rules that gives us certainty in our daily lives- be it the certainty of what conduct is permissible on the road or what rights we have under an

**Data Availability Statement:** All relevant data are within the paper and its Supporting Information files.

**Funding:** The author(s) received no specific funding for this work.

**Competing interests:** The authors have declared that no competing interests exist.

employment contract. If the law is the framework of our society, lawyers themselves serve as guides, helping us navigate life's challenges.

The COVID-19 pandemic caused an unprecedented disruption to all of society. The practice of law was no different. As businesses shut down and masks and social distancing became part of the common lexicon, the judicial system similarly struggled to adapt to new norms. During 2020 these norms changed seemingly daily, as more was learned about the disease and as policy makers worked to balance economic well-being with public safety. The most apparent disruption was a general suspension of jury proceedings and live court in general. Like every other aspect of society, lawyers were forced to adapt to new conditions of work on a day-to-day basis including video conferencing with judges, courthouse staff, clients and colleagues. Depositions were taken by video and the court system found ways to continue its work despite the challenges posed by COVID-19. This study looks at the ways that these disruptions resulted in decreases in work satisfaction as well as increases in adverse wellness impacts such as stress, alcohol use, and mental health issues.

These disruptions were overlaid onto a legal profession already long-recognized as being filled with stressors. The extant research indicates attorneys typically experience more stress and depression, and show greater incidence of alcoholism, substance abuse, and depression than the general population [1]. The practice of law for many involves the internalization and assumption of the burdens that others face in their daily lives. For everyone, the stresses of 2020 were beyond the norm. For Kentucky's Bar in particular, the stresses were pronounced. In the midst of the pandemic, the profession was rocked by a series of attorney suicides. Because of confidentiality, exact numbers were difficult to gather. The Bar Association cannot always know, nor disclose even when known, that attorneys have died (much less attempted) suicide. In a three week period from Christmas 2020 until mid-January 2021, at least four Kentucky attorneys died by suicide. Louisville lawyer Wilson Greene, summarized the views of many of his colleagues, reporting that attorneys are afraid to get help or go into treatment for fear of losing referrals from other lawyers. He noted that "lawyers are also problem solvers and fixers," and can spread themselves too thin. "The legal profession is also adversarial. That sets up a win/loss situation" said Greene. "Your clients are looking for positive outcomes and sometimes that doesn't occur. That's unfortunately a reality of the practice of law. But that drains you, and can pull you down" [2].

Regardless of the exact numbers, for many attorneys, it appeared that the 2020 pandemic had pushed their stress past the breaking point. This study aims to illuminate in more detail the impact of the pandemic on the well-being of practicing lawyers. Consequences of balancing work and life under pandemic restrictions and uncertainty is expected to include decreased mental health, increased substance use, and challenges with maintaining client, colleague and family relationships due to the pandemic. The implications of this could have serious consequences for the public who need legal counsel, services and support.

This paper begins by reviewing the relevant past literature with regard to well-being as related to the legal profession in general. Next, we discuss the methodology of this unique survey, whose origins lie in the efforts of the Kentucky Bar Association to adapt to the 2020 pandemic. The survey itself covered over seventy inventory items. This paper briefly discusses the items key to this inquiry and outlines the wellness measures used. We then discuss the results related to wellness measures such as stress, relationships, substance and alcohol use, and work satisfaction, which showed negative impacts from the COVID-19 pandemic to varying degrees throughout the bar membership. Finally, the paper concludes by looking at recommendations to address the issues identified, including discussion of some steps undertaken by the Bar itself.

## Literature review

The Centers for Disease Control and Prevention (CDC) defines health related quality of life in terms of an individual's perception of their physical and mental health or overall well-being [3]. Though there is not one agreed upon measure for well-being, studies often relate a person's well-being to their physical and mental health, substance use, stress levels, and job and relationship satisfaction [3, 4]. The American Psychological Association [5] includes healthy relationships, lack of anxiety, and an ability to cope with daily activities as part of emotional well-being. The annual National Survey on Drug Use and Health [6] is a well-established indicator of general population health and well-being. NSDUH trends between 2008 and 2018 include increased illicit and prescription drug misuse (17.8% to 20.8%) and increased mental illness diagnoses (17.7% to 20.6%). Major depressive episodes diagnosed among 7.8% of the population were up from 6.6% a decade earlier. There was also an increase in adults reporting suicidal thoughts or ideation (3.7% to 4.8%).

Another measure of general population health is the Household Pulse Survey which collected its most recent dataset during the pandemic between August 2020 and February 2021. These HPS data reflect increased past week symptoms of anxiety or depression (36.4% to 41.5%), though accessing the counseling or therapy needed over this time period was less likely to happen than pre-pandemic [7]. Specific to legal practice, Somers and Casal [8] examined coping among lawyers during the pandemic and found the most frequent coping techniques were dealing with stress on their own and/or using drugs or alcohol to cope.

The psychological well-being of lawyers has been a serious concern for the profession for several years now due to alarmingly high suicide rates, drug overdoses, and reports of severe depression [9–12]. Over 67% of lawyers report regularly working more than a typical 40-hour week [13] with one study reflecting 15.3% of lawyers working over 60 hours a week on average [14]. In addition to work-load and hours, lawyers report high job-related stress levels, anxiety, and risky drinking behaviors [9, 13].

An extensive 2016 study conducted by Krill and colleagues examined the prevalence of mental health and substance use among a sample of 12,825 practicing lawyers. The majority reported drinking alcohol in the past 12 months (84.1%). Scores on the AUDIT, a standard alcohol use measurement tool, indicated 20.6% of participants had problematic drinking patterns and 36.4% met diagnostic criteria for hazardous drinking or possible alcohol abuse/dependence. Other substance use reported by lawyers included tobacco (16.9%), sedatives (15.7%), marijuana (10.2%), opioids (5.6%), and stimulants including cocaine (5.6%). Depression, anxiety and stress levels were self-reported using the DASS-21 scale. Mild to moderate depression levels were found among 19.9% of lawyers and at severe levels among 8.4%. Anxiety at the mild to moderate level was reported by 13.6% and at the severe level by 5.6%. Mild to moderate stress levels were reported by 17.0% of lawyers and at a severe stress level by 5.7%. When participants were asked about barriers to obtaining support services to cope with depression, anxiety and/or stress, the main concern was regarding privacy and not wanting other professionals or their clients to know they needed assistance [13].

Specific to law practice, Bloomberg Law's Attorney Workload and Hours Survey captured data during the first year of the pandemic which reflected no reduction in typical work hours due to the additional pandemic challenges [15]. The survey found a decrease in lawyer job satisfaction and a decline in overall well-being [15]. In general lawyers during the pandemic have reported 3 to 4 times more anxiety and depression than the general population [16], yet they are unlikely to seek treatment or support.

Lawyers undertake a unique calling wearing many hats. It may be this unique, shared experience that makes lawyers more susceptible to wellness issues. They are advisors, providing

technical legal guidance on often arcane subjects. They are advocates, being called upon to fight for their client's interests. They are also counselors. While not formally trained as such, many lawyers find themselves assisting individuals at the most difficult times of their lives. As such, they cannot be mere technocrats and must be empathetic. The best of them will navigate emotion as skillfully as the technical law. Small firm or solo practitioners bear the added burden of being themselves small business owners, with the attendant responsibilities and stressors coming from that role as well. With this in mind, it is little surprise the wellness of lawyers has historically been at risk.

Understanding the impact of the pandemic on the overall well-being of lawyers is important to the client's they serve and the professionals who work with them. Lawyers are essential to the functioning of our society and legal system. Thus, there is a need to maintain policies and protocols that ensure ongoing support for lawyer well-being in a profession that is historically high stress [17]. The research study described in this paper examines the experiences of lawyers in regard to their well-being during the pandemic. This paper also considers steps already taken by one state's Bar to increase attorney access to mental health and psychological supports.

## Methods

The survey originated from the desire of the author, a member of the Kentucky Bar and part-time practitioner, to first understand the shared experience of these disruptions and second, to create a historic record of the legal practice during this unique time. The survey was drafted in consultation with various stakeholder groups in Kentucky's justice system including the Bar Association and the state's Administrative Office of the Courts, the administrative agency reporting to the Commonwealth's Supreme Court that oversees all courts within the state. The data collection covered a broad range of disparate topics including access to justice issues, the financial impact on attorneys and their communities, collegiality, and a large inventory of questions regarding the mechanics of practicing in a socially-distanced, online environment. The instrument also consisted of inquiries regarding attorney wellness. This wellness inventory is the focus of the present paper.

### Ethics

The Institutional Review Board at Eastern Kentucky University, the authors' institution, approved this project as an "exempt," activity. An initial disclosure statement was provided as the first page of the online survey and participants provided affirmative informed written consent before moving forward. The informed consent appeared as an introduction page on the survey instrument and read as follows:

### KY bar practice in pandemic survey

You are being invited to take part in a research study on the impact of the pandemic on the practice of law in Kentucky. This study is being conducted by Michael Fore at Eastern Kentucky University in conjunction with the Kentucky Bar Association. Survey results will be used to understand how the pandemic has impacted the practice of law across the state and may also be used by the KBA to direct member resources and develop programs.

If you decide to participate in the study, you will be asked to complete a survey about your practice experience during the 2020 pandemic. Your participation is expected to take no more than 15 minutes.

This study is anonymous. You will not be asked to provide your name or other identifying information as part of the study. No one, not even members of the research team or the KBA,

will know that the information you give came from you. Your information will be combined with information from other people taking part in the study. When we write up the results of the study, we will write about this combined information.

We will make every effort to safeguard your data, but as with anything online, we cannot guarantee the security of data obtained via the Internet. Third-party applications used in this study may have terms of service and privacy policies outside the control of Eastern Kentucky University.

If you decide to take part in the study, it should be because you really want to volunteer. You will not lose any benefits or rights you would normally have if you choose not to volunteer. You can stop at any time during the study and still keep the benefits and rights you had before volunteering.

This study has been reviewed and approved for exemption by the Institutional Review Board at Eastern Kentucky University as research protocol number 003718. If you have any questions about the study, please contact Michael Fore at michael.fore@eku.edu. If you have questions about your rights as a research volunteer, please contact the Division of Sponsored Programs at Eastern Kentucky University by calling 859-622-3636.

By completing the activity that begins on the next screen, you agree that you (1) are at least 18 years of age; (2) have read and understand the information above; and (3) voluntarily agree to participate in this study.

Consented was accepted by clicking to continue the survey, providing written consent to participate. All responses were anonymous, and names were not linked in any way to responses. The Qualtrics survey software collects longitude/latitude data as well as unique IP addresses, thus respondents were able to leave and return to the survey to complete their responses.

In terms of confidentiality, attorneys are bound by an ethical duty to report violations of ethical misconduct by themselves or others under existing professional regulations (Kentucky Rules of Supreme Court, SCR 3.130(8.3)). Attorneys/survey respondents would also be already prohibited from disclosure of confidential information (Kentucky Rules of Supreme Court, SCR 3.130(1.6)) Upon conclusion, the anonymous survey results were, by agreement with the Kentucky Bar Association, made available to all bar members via the organization's web site (see https://www.kybar.org/page/pandemicsurvey2021).

The survey itself ran from February 8 until February 12, 2021 and was sent to all Kentucky Bar members. By rule, all members of the Kentucky Bar are required to maintain a valid email address for official communications (Kentucky Rules of Supreme Court, SCR 3.035). The first email contact was part of the Bar's regular periodic newsletter while subsequent emails came out as stand-alone requests for survey participation.

## Measures

**Well-being.** As there is not currently a standard measure of well-being, questions focused on the most common indicators of health, positive relationships, and access to behavioral health resources. The survey measured well-being through self-reported responses to questions that asked about the following: stress of law practice; work satisfaction ratings; relationships with clients, colleagues, and family; depression; alcohol and cigarette use; and exercise. Participants were asked about their perception of access to resources to handle stress, substance abuse, and mental health problems. When appropriate, participants were asked to rate experiences pre-pandemic and during the pandemic or the present time period (February 2021).

**Demographics.** The characteristics used as background demographics for the survey including time in practice and practice area categories are the same characteristics the Bar

routinely uses to track its membership. The Kentucky Bar does not routinely track its membership by characteristics such as race, disability, and sexual orientation. As such, these characteristics were not measured as part of this survey. While measurement of the impact of the pandemic on attorneys by these characteristics certainly warrants some study, this is not the focus of the current work.

The study included measures to examine size of practice and location. Size of practice may have a relationship with well-being measures in relation to accessibility to resources and support. A variety of factors have historically favored lone attorneys, particularly in more rural parts of the state. For example, the state's civil and criminal courts, property indexing, and other judicial functions are organized on a county-by-county basis. The one hundred and twenty (120) counties that make up Kentucky are unusually small compared to most other states. Historically, the stated justification was that a county's borders should stretch no further than a round-trip horseback ride to the county seat. Thus, despite being a relatively small state, Kentucky has the 4th highest number of counties of any state in the nation.

Location of practice was also measured as it may have implications for respondent well-being. Each of the one hundred and twenty counties has its own county seat, its own Courts of various jurisdictions, its own property records, etc. Because of this dynamic, the practice of law in Kentucky has traditionally been more localized. Each courthouse in each county supported its own, in many cases insular, ecosystem of legal business [18, 19]. This meant that the smallest and least populous community usually had access to local legal counsel. This also meant that for more isolated, low-population areas, there was less economic incentive for larger practice groups to develop. Indeed, many rural Kentucky counties might traditionally have only had a handful of attorneys practicing at any given time. The limited resources and generally less prosperous economies of these rural areas simply did not provide enough economic "oxygen" for the development of multi-attorney practice forms seen in Kentucky's cities. However, in recent years as the internet has made the world relatively smaller and transportation barriers have shrunk, the small-town practice of law seems to be fading in favor of larger multi-county firms.

Because of this unique characteristic, Kentucky law firms tend to be smaller and the traditional definitions of "small" and "mid-sized" firms used by national organizations do not always fit the reality of Kentucky's fractured legal landscape. For example, a firm of 5 attorneys would be considered small by the measures of the American Bar Association but may well be a larger firm in any Kentucky County outside of Lexington, Louisville, or the Cincinnati suburbs of Northern Kentucky. Measures for this study were defined as solo practice (1 attorney), small to mid-sized practices (2–20 attorneys), and large practices (over 20 attorneys). While these definitions might not fit a more urban area, they are a good measure for Kentucky's more decentralized Bar.

The survey also considered community type in which the attorney practiced–rural, suburban, or urban. According to U.S. Census definitions, Kentucky is primarily a rural state [20]. Outside of the urban areas of Louisville and the Cincinnati suburbs of Northern Kentucky, the state's largest urban area is Lexington, a city of just over three hundred thousand surrounded by famous horse farms [21]. Over half of the Commonwealth's residents live in urban areas meaning rural areas are less densely populated [22]. Again, the classification of rural versus urban used in this survey might not have been well-suited to a more urban state, but do capture the essence of the practice experiences for Kentucky. Outside of the state's three urban zones (Lexington, Louisville, and the Northern Kentucky suburbs of Cincinnati), the state is largely rural. The state's large number of counties has historically slowed the development of more regional centers, as each of the 120 county seats has its own distinct legal network.

**Qualitative comments.**   In addition to the topics discussed previously, the instrument also included open-ended inquiries that gave respondents the opportunity to comment on a variety of practice issues. All told, the survey took on average 10–20 minutes to complete. This paper focuses solely on queries relevant to lawyer well-being and quality of life. Each question is analyzed based on all respondents who answered that question.

## Results

### Demographics

Over two thousand three hundred (2,311) members of the Bar took part in the anonymous online survey. Members provided candid, thoughtful responses on a wide range of topics related to the pandemic's impact, access to justice, collegiality, and quality of life. The membership responded at a very high rate for an online survey, with over eight percent (8%) of total statewide membership participating. Respondents by length of time in practice closely reflected the Bar's overall make-up and distribution (See Table 1).

Like most other state Bar Associations, Kentucky's Bar skews older as a result of many factors. For example, changing Bar admissions rates, bar exam passage rates, and law school access, admission, and passage rates all restrict to a relatively fixed annual number of newly licensed attorneys. Secondly, as the baby-boom generation ages, the overall American employment demographics have trended older. Further, the practice of law itself sees a fairly high attrition in early years of practice. By the very nature of the work and legal market, many would-be practitioners are weeded out early by economic factors or self-select away from the stresses of practice. However, once that initial attrition period has passed, individuals that continue in beyond five years tend to remain in practice for a long time. After the steep learning curve of practice is passed and the initial economic pressures stabilize, many remain in the profession beyond what might be a typical retirement age. The practice of law is not a vocation generally limited by physical age. Unlike other vocations, there are almost no physical barriers to continued practice of law well past traditional retirement ages.

Because the survey was developed and implemented primarily to aid the Bar and Kentucky's Courts in coping with the pandemic, the survey itself was focused on different demographic factors than might typically be utilized in this type of research. The Bar itself primarily classifies its membership by practice setting and length of time in practice. Length in practice is particularly important for the Bar Association. The practice of law has a steep learning curve. In recognition of this and the traditionally high attrition rate of new lawyers, the Bar provides additional resources, such as trainings and legal education courses, particularly

**Table 1.  Years of practice comparing general bar membership to survey respondents.**

| Years of Practice | Bar Membership (n = 19,296) | Survey Respondents (n = 2,262)* |
|---|---|---|
|  | % | % |
| 2 years or less | 6.5 | 4.0 |
| 3–5 years | 6.8 | 6.8 |
| 6–10 years | 13.9 | 11.6 |
| 11–15 years | 12.7 | 13.4 |
| 16–20 years | 10.4 | 11.1 |
| 21–25 years | 10.5 | 12.4 |
| Over 25 years | 39.0 | 40.6 |

*49 missing responses

focusing on new and younger attorneys in an attempt to ease their acclimation to the practice of law.

Attorneys from all fields of practice from all areas of the Commonwealth participated in the survey (See Table 2). Respondents provided professional demographic responses including their time in the practice of law, their field of practice, and the community setting in which they operate. The quarter of respondents practicing in solo settings is consistent with Kentucky's historical geography.

Over half of the Commonwealth's residents live in urban areas and the survey respondents reflected this mix (Kentucky League of Cities, 2021). Of the 2,232 lawyers who responded to this question, over half (55.2%) reported practicing in urban areas, 28.3% in a small city or suburban area, and 16.6% practiced in a rural community.

**Well-being measures.** The COVID-19 pandemic emotionally-taxed the entire world. Not surprisingly, Kentucky's attorneys were no exception. Because of the already well-recognized stressful and emotionally difficult aspects of the profession however, this impact on well-being deserves additional scrutiny. Measures for well-being included stress of practice, work satisfaction, mental health, substance and alcohol use.

**Stress of practice.** The majority of attorneys agreed that law practice has become more stressful (74.4%) during the pandemic and more difficult (76.9%). By practice type (i.e., solo, small-mid size firm, large firm, prosecutor, public defender/DPA, all other) more public defender/DPAs reported increased stress with their practice during the pandemic, (85.1%, $X^2(10, 1689) = 19.35, p = .036$).

Some respondents identified specific overflow from work conditions translating into personal stress. For example:

> *We as family law attorneys are dealing with families extremely stressed over COVID. Marriages/families are stressed when dealing with normal day to day activities. Add loss of jobs, school closings, and people being together 24/7 and you have the recipe for disaster in many families. Also, when children are not in school, there is no independent person keeping an eye on them for physical abuse or psychological problems. Victims of domestic violence are more dependent upon the abuser and may have less avenues of support outside the home, such as family, friends, or community services.*

The majority of lawyers felt they had someone they could talk to about their stress (88.3%) and knew of resources available to help them with their stress (81.1%). On the other hand, by practice location significantly more rural respondents (25.0%) reported being unsure of resources to help them with stress compared to urban respondents (17.6%) $X^2(1, 1485) = 7.125, p = .008$. Among different attorney types, criminal lawyers were least likely (27.2%) to

**Table 2. Primary practice setting of respondents (n = 2,272).**

| Practice Setting | n | % |
|---|---|---|
| Solo Practice | 630 | 27.7 |
| Small Firm (2–5 Attorneys) | 497 | 21.9 |
| Mid-size Firm (6–20 Attorneys) | 253 | 11.1 |
| Large Firm (over 20 Attorneys) | 250 | 11.0 |
| Public Defender, DPA, prosecution, other | 579 | 25.5 |
| Judicial | 63 | 2.8 |

*39 missing responses

believe resources were available to help with stress, $X^2$ (2, 1479) = 12.22, $p$ = .002. To highlight the issues, one respondent stated:

> *My practice has always been relationship intense. I think the practice loses something when we can't see our clients face to face. I feel more connected to clients and their problems when I actually meet with them in person. My office is still very efficient, but helping people needs to more than the appropriate filing of documents, the meeting of deadlines, and sending the appropriate amounts of correspondence. The joy of practicing law also depends on meeting with clients, seeing their facial expressions, and even receiving the occasional hug from a sad or grateful client.*

**Work satisfaction.**   Median ratings of work satisfaction on the scale from 0 = least satisfied to 100 = most satisfied were higher before the pandemic (79.09) compared to current ratings (62.95). Like much of the global economy, the legal system suffered disrupted operations during the pandemic. The biggest dip in ratings occurred for judges and those in solo practices whose job satisfaction ratings went down by about 19 points. (See Table 3).

Deeper analysis shows job satisfaction was also related to years of practice among attorneys. ANOVA comparing average satisfaction ratings before the pandemic to the present revealed satisfaction fell the most among attorneys with two or fewer years of experience who had an average satisfaction rating 22.42 points lower than any other age group, $F$(6, 1964) = 3.75, $p$ = < .001. In addition to ratings, attorneys were asked about the overall impact of the pandemic on their law practice ranging from very positive, positive, neutral, negative, or very negative. Negative impact was greater the more years an attorney had practiced, though over 60% of all groups reported a negative effect from the pandemic: < than 5 years 61.9%; 6–15 years 64.1%; 16–25 years 63.8%; and over 25 years 65.6%.

On the whole respondents agreed that law practice has become more difficult and more stressful during the pandemic with varying impacts noted on family life, mental health, and relationships with colleagues. One respondent highlighted the difficulties by saying:

> *The practice of law is not sustainable. I have serious concerns that this is not a profession that a person can or should practice for a long period of time. It demands too much, the stress is too high. It will end up killing or maiming many, if not all who make it a life's work.*

**Relationships.**   Over half (58.3%) of respondents agreed that the pandemic had made it hard to maintain positive relationships with other attorneys or that (57.2%) "the demands of their practice interfered with my home and family life." Forty-six and a half percent (46.4%) agreed engaging in the practice of law "put a strain on their family," while 64.3% brought stress home with them during the pandemic.

**Table 3. Work satisfaction median ratings before pandemic to present by practice types.**

| Practice Type | Median Work Satisfaction Ratings (Scale 0–100) | | |
|---|---|---|---|
| | **Before Pandemic** | **At Present** | **Difference** |
| Solo Practice | 79.42 | 60.59 | -18.83 |
| Small to Mid-size Firm | 75.54 | 64.71 | -10.83 |
| Large Firm | 79.15 | 70.09 | -9.06 |
| Public Defender, DPA, Prosecution, Other | 75.24 | 65.45 | -9.79 |
| Judicial | 81.00 | 62.00 | -19.00 |

Specifically, those in a large firm (65.8%) or who served as public defenders/DPA attorneys (66.0%) were most likely to indicate challenges keeping up good relationships with other attorneys, $X^2$ (10,1688) = 23.46, $p$ = .009. Public defenders/DPA attorneys (78.7%) were most likely to report bringing stress home with them, $X^2$(10, 1685) = 28.98, $p$ = .001. They also felt their law practice put more of a strain on their family during pandemic than other attorneys (56.5%), $X^2$(10,1685) = 26.86), $p$ = .003. Attorneys from large firms (70.9%) were most likely to feel the demands of work interfering within their home life, $X^2$(10,1688) = 37.18, p < .001.

As one respondent stated:

*The impact on the volume of cases we now have for the poor, elderly, and domestic violence victims makes it even more difficult to balance work and personal life. Although it has not affected me in this particular area, I do know colleagues who have suffered greatly financially as a result of clients losing their jobs, etc.*

**Mental health.** Over a quarter (28.4%) of lawyers reported experiencing frequent depression during the pandemic. By practice location, more rural (29.3%) than urban (27.9%) respondents indicated frequent depression, $X^2$(2,1666) = 6.66, $p$ = .036. By type of attorney, the relationship with frequent depression was greatest for public defenders/DPA attorneys (48.9%), $X^2$(10, 1685) = 20.05, $p$ = .029.

Approximately half (50.4%) of respondents indicated they were aware of other attorneys who were dealing with mental health issues. Public defenders/DPA attorneys (72.1%) reported this awareness significantly more than other types of lawyers, $X^2$(5, 1576) = 15.67, $p$ = .008.

In regard to physical exercise which aids in managing mental health, 42.8% of lawyers agreed they exercised less now than before the pandemic. There were no differences between groups in decreased exercise indicating it was an issue across the board. One respondent put it this way:

*It has become a lonely profession. I do not know what some of my clients look like, the clerks in family court are on overload. The Judges are trying to do the best they can, but a few of our ten family court judges take out their stress of the job on attorneys and do so in the most unprofessional way-publicly. The pandemic has made that worse. I have witnessed a judge call an attorney stupid in the presence of their client. The pandemic has simply worsened that sour attitude.*

**Substance and alcohol use.** One quarter of respondents (25.0%) indicated they were drinking more alcohol currently than before the pandemic. By practice location, significantly more urban (26.1%) than rural respondents (19.0%) reported increased drinking, $X^2$(2,1667) = 10.57, p = .005. Practice type was significantly associated with increased drinking for attorneys from large firms (35.2%), $X^2$(10, 1686) = 27.08, $p$ = .003.

Four in ten (41.4%) respondents indicated they were aware of other attorneys dealing with alcohol or substance use issues. By type of attorney, public defenders/DPA attorneys (62.2%) were more likely to indicate this knowledge of substance use issues, $X^2$(5,1591) = 11.90, $p$ = .036.

## Discussion

### Stress of law practice

Increased difficulty with practice was noted by over three-fourths of all respondents, though public defenders/DPA attorneys seem to have had the most difficulty. These criminal defense

lawyers reported a higher incidence of stress beyond the increases noted in other practice areas. The specific reasons for this increased stress among public defenders/DPA attorneys can likely be traced to the way the pandemic impacted criminal practice. For much of 2020, jury trials were cancelled due to COVID restrictions. Without the threat of a jury trial (and potential convictions), there was little incentive to make plea agreements. Without plea agreements or trials to bring criminal prosecutions to a conclusion, cases never ended. As a result, workloads grew throughout 2020 as attorneys, particularly public defenders, saw their workloads expand exponentially. There was no way to predict plans for the future as the pandemic continued to shift and change.

Further, the daily nature of criminal defense work was particularly complicated by the move to virtual court. Unlike areas of civil trial work which are typically more planned and set-piece, criminal practice requires the ability to advise clients in real-time. Defendants need to be informed about and explained things like plea offers and receive guidance as to testimony or when to invoke the 5th Amendment and remain silent. In an in-person court proceeding, these talks would happen as quick asides between attorney and client. Zoom, the frequently-used videoconference platform, has no meaningful option for truly private, quick, and confidential communications during the midst of an online court proceeding. Further, criminal defense attorneys need open and honest communications with clients to be able to provide good advice. All of these practices were impaired or made impossible by the move to virtual court hearings. So not only did criminal defense workloads soar under cases which never ended but they became unable to perform their most important work in the virtual settings.

## Work satisfaction

Younger practitioners had the greatest dip in work satisfaction ratings. This group typically has less professional agency- they have less choice of work, receive lower salaries, and may be required to undertake less desirable work thus the impact of the pandemic may have hit them the hardest. It is unclear what this may mean for the legal profession in the long-term. Attrition is already a serious problem in the early years of a legal career. If this decreased satisfaction increases this existing problem, then it may well translate into a loss in the availability of legal service providers in years to come. Similarly, younger attorneys may be disinclined to actually seek assistance, regardless of the psychological resources available. Particularly for those practicing in large firm settings, employment is typically competitive. Young attorneys there must seek to rise in the organization and earn performance bonuses. These demands often create an environment where the stigma of perceived "weakness" would discourage those in need from seeking assistance. If they are part of a solo practice, they may have fewer resources and less leeway in managing their work than larger firms or public agencies. The younger attorneys likely have less knowledge of how to access additional help and are still trying to figure out how best to serve their clients which was complicated by the restrictions in practice and client access.

Judges also had a significant dip in satisfaction. For many, joining the bench is already a solitary calling. Whereas attorneys typically socialize with other attorneys regularly, by the very nature of the job a judicial post is somewhat more isolated. Take for example, litigation attorneys. During traditional times, they communicate with their colleagues daily as part of the normal adversarial process. Attorney communication with their colleagues is a necessary part of the process from the routine- scheduling matters, discussing pending motions and issues in discovery, to the more significant such as confidential settlement negotiations. Judges have none of this regular peer-to-peer interaction. To the contrary, members of the judiciary are expected to be neutral and avoid the appearance of any favoritism. This means they must

actively avoid the routine, regular communications with other attorneys that are daily fare for most attorneys. It is these regular communications that build the collegial foundation of the profession and often serve as stress relievers. Judges must necessarily avoid this. As a result, their only regular social interaction with other attorneys is at the actual court hearings. The pandemic has moved these proceedings online making an already lonely calling on the bench even more so.

## Relationships

The practice of law is built on relationships. Despite adversarial positions in litigation or other dealings, much of the practice of law requires good communication and "professional courtesy" of fellow practitioners. From navigating deadlines, to scheduling, to negotiations, much in a lawyer's day is spent navigating the system cooperatively with their co or opposing counsel. The pandemic necessitated distancing and moved much of this communication to the virtual world. This move appears to have negatively impacted relationships between colleagues.

Familial relationships should be a source of comfort and stress relief. Home life should be kept separate if possible, from work life in order to allow time to decompress and enjoy other pleasures or hobbies. Many attorneys had to work from home and took on additional tasks to try to keep up with the challenges of the pandemic and its restrictions on work. Without the clear delineation between work and home life, the stress on relationships seems to have exponentially increased for attorneys.

## Mental health and substance use

Specific aspects of the emotional toll also raised cause for concern including increased reports of frequent depression. Half of respondents knew someone with mental health concerns, yet there was a lack of awareness of resource access that seems particularly problematic for rural attorneys and public defenders/DPA attorneys. Rural communities historically have fewer resource options and less ability to seek mental health counseling without others in the community finding out about it. Even parking in the mental health center lot could easily be witnessed by clients or fellow attorneys thus stigmatizing the help seeking attorney. In general, seeking mental health care is often perceived as admitting weakness. Attorneys who appear "weak" may lose clients or even their jobs due to the stigma. As we see the increased suicide rates among attorneys, it becomes even more important to destigmatize mental health care for depression and anxiety.

No doubt the legal profession was not alone in drinking more during the massive disruptions of 2020. However, given the legal profession's traditional struggles with alcohol abuse which can start in law school, this uptick can't merely be written-off as part of a larger trend. Large firms in particular indicated more of a problem with increased drinking and 4 in 10 attorneys reported knowing of other attorneys with substance use problems. Again, the stigma of seeking help could be a major issue for many attorneys who view seeking support to manage drinking problems as a weakness. Though the majority of respondents felt they had someone to talk to about their stress, this likely does not mean they are talking with a licensed professional counselor or psychologist. Providing resources specific to the challenges faced by attorneys in their multi-faceted support roles with clients could be beneficial.

## Conclusions & recommendations

The legal profession plays a unique role in civil society. From protecting the rights of defendants to guiding businesses, the work of legal practitioners has a wide-ranging impact. Similarly, the impact of the COVID pandemic on society has and continues to be wide-ranging. As

this study reveals, the legal profession was not immune from COVID's impacts. The COVID pandemic had a negative impact on numerous aspects of attorney well-being as well as their ability to practice law effectively and to be available to the public who need their counsel.

Because of the unique role that lawyers play in society, attorney wellness has an impact beyond the legal profession. Attorneys who are impaired, depressed, or struggling with substance use may have social implications for their larger communities. The impaired or unwell attorney may miss deadlines, be unavailable for client contact, or otherwise be compromised in their ability to assist others in their times of need. In short, the impact of lawyer wellness stretches far beyond the legal system and has impacts for society as a whole.

Just as the stress and wellness problems that COVID exacerbated do not have a single cause, there is similarly no one solution to the legal profession's wide-ranging personal wellness problems. Every state has some type of lawyer assistance program (LAP) that offers confidential assistance for attorneys to connect them with licensed professionals for counseling or treatment when needed. Each state's LAP is different, but one group working to support attorney wellness in a comprehensive and effective way is Kentucky's Bar Association (KBA). The KBA worked to originate the present study and is already using survey findings to redirect its resources to address these changing problems.

## Kentucky's lawyer's assistance program

KBA has long recognized the challenges to attorney wellness and the need to reduce stigma for seeking counseling or treatment. In 2003, Kentucky's Supreme Court founded the Kentucky Lawyer's Assistance Program (KYLAP) by adoption of Kentucky Supreme Court Rules 3.9000 to 3.995 [23]. KYLAP is a program with a wide mandate to address any issues which might impair a lawyer's ability to perform their job. It exists to provide Bar members with resources to address a variety of problems including addiction and psychological or emotional illness that may impair a person's ability to practice law or serve on the bench.

Rule 3.910 defines generally KYLAP's mandate:

> . . .The types of assistance offered and provided by KYLAP in a particular case may include lay counseling and encouragement; assisting, planning and execution of interventions; providing information about treatment alternatives; monitoring progress of recovery from impairment, which may include assistance in arranging, scheduling and tracking attendance at recovery programs, appointments with counselors, therapists and medical care providers and compliance with alcohol or drug screens; monitoring compliance with voluntary or involuntary treatment or recovery programs, which may include documentation and reports concerning compliance or non-compliance; obtaining authorizations in conformity with federal and state law; and other related tasks that may assist a member of the said community in addressing an actual or potential impairment; provided, however, that KYLAP shall perform the aforesaid types of assistance in such a manner that KYLAP's staff does not render legal or medical advice and does not engage in any activity which constitutes the practice of law or medicine. (KY. Supreme Court Rule 3.910 (2)).

KYLAP primarily acts as a referral service connecting in-need Bar Members with a variety of mental health and substance abuse resources (https://www.kylap.org/resources/). The organization also provides peer support, mentoring, and support groups of similarly-situated lawyers. In the wake of the COVID pandemic and based on the survey findings, the Bar has redoubled its effort to inform attorneys about these resources. Notably, unlike many assistance programs in other areas of society, KYLAP is triggered by a lawyer's impairment to work and

perform the duties of their job, not necessarily the needs of the individual. Impairment is defined as:

> . . .any mental, psychological or emotional condition *that impairs or may foreseeably impair a person's ability to practice law or serve on the bench*. Impairment may result from addiction to intoxicants or drugs, chemical dependency, substance abuse, mental disease, mental disorder or defect, or psychological or emotional illness. (KY. Supreme Court Rule 3.910 (2)).

Recognizing the unique role of the legal profession, the program not only exists to meet the needs of Bar members but exists also to protect those dependent on the troubled lawyer for their own well-being. Unlike other vocations, impaired attorneys pose a risk not just to themselves, but to their clients. Clients depend on their attorneys for counsel, guidance, and protection of their own rights.

Confidentiality is a cornerstone of KYLAP services. (Ky. Supreme Court Rule 3.990). Disclosure of impairment would certainly deter lawyers from seeking help. Public knowledge of an impairment would certainly prevent an attorney from continuing to be effective in an adversarial setting, such as litigation, even after the problem was resolved. It is the unfortunate fact that the mere knowledge of a past problem might stigmatize the attorney and lead to them being treated differently by colleagues, judges, or opponents. It is also easy to imagine that a lawyer with known mental health or substance use issues would find few clients seeking their services. Confidentiality is however a double-edge sword. In both professional counseling settings and the KYLAP mandates, confidentiality is established to protect the attorney. On the other hand, this makes it difficult to assess the effectiveness or utilization of services among attorneys. Stigma around seeking counseling or treatment is perpetuated through confidentiality making it hard for attorneys to share with each other that they need help or are receiving help.

However, while KYLAP is able to offer resources to help members deal with the stress of law practice, there are limits to the program. One respondent noted:

> It would be nice if lawyers on the verge of a nervous breakdown could call KYLAP and have them intercede with a judge on our behalf to continue a case for 30 days or something.

The program has no authority to actually intervene in judicial proceedings. While KYLAP can offer resources to deal with stress, it has no authority to address the root causes of attorney stress- the work itself. The judicial system could scarcely function without stress. By its very nature, litigation work in particular inherently involves conflict. The system exists to resolve disputes and some stress is endemic in the process. Further, the judicial system exists to serve its constituents- clients, litigants, and society as a whole. Lawyers have an important role in the operation of the system but aren't, and shouldn't be, the focus.

## Social media

For many across the world, online social media platforms offered a way to connect outside of the home during the pandemic lockdowns. Social media also offered meaningful connections for some survey respondents. One solo practitioner discussed this approach to counter pandemic isolation:

> We know that the law is isolating to begin with, and with the pandemic, that has become more evident. I, as a solo, am thankful for attorney and KBA Facebook groups because I do not feel alone that way.

Increasing online support groups that are anonymous yet aimed specifically at the needs of attorneys with mental health or substance use issues could benefit the community. The connections available through online telehealth services are an important part of healthcare particularly in rural communities and they have been expanded into urban regions as well as for counseling purposes. Telehealth is confidential and doesn't require driving to an office to meet with a counselor which reduces the possibility of being seen in a community and potentially stigmatized by clients or colleagues. Expansion of these types of online services particularly for attorneys could be beneficial.

## Learning self-care strategies

*I am happier now than I was in March, 2020. Being able to work from home and having a slower pace helped me substantially. However, I already had established self-care strategies to manage my own mental and physical health and seek support where needed. . .*

Development and utilization of self-care strategies is important to anyone in managing stress. Lawyers are no different. Nor does their work as counselors to clients necessarily mean they are any better suited to deal with life's difficulties than anyone else. Further, theirs is a competitive and stressful profession which has not embraced concepts of self-care. The best lawyers are often held up and revered for their client-centric approach and long hours at work, as opposed to their success at achieving work-life balance or a good quality of life.

There is an opportunity here for psychologists and therapists to reach out to the attorney community and connect around prevention and coping skills. The Bar associations have opened the door and created an avenue for engaging with attorneys to provide these kinds of educational programs. The subject of well-being needs to be discussed openly and regularly to reduce stigma and normalize reaching out for help when the stress of work threatens to overwhelm us.

*We need to begin substantively addressing mental health issues in attorneys—beyond just telling them to meditate and take a hot bath (which is the advice I received at the last stress management seminar I went to during KBA). Expectations for the practice of law—especially for mothers—need to be radically adjusted. There needs to be a change to the entire culture. It is an enormous task that will likely take decades but it needs to be started and supported by people at the top—judges and the heads of large firms especially.*

At some level, the practice of law inherently includes aspects of work that are adversarial. Some legal work is, of its nature, a zero sum game. Criminal defendants will either be convicted or not, personal injury plaintiffs will either be compensated for their injuries or not. In such an environment, some degree of heightened stress is unavoidable. However, a conversation should take place among the Bar about how members can engage in this work while preserving their personal health. This conversation should include the psychologists and other professionals who provide training on coping with stress. Knowing these same professionals can also be there to extend a hand if an attorney needs additional help with addressing mental health or substance abuse issues. By focusing on collaboration between the Bar and helping professionals, a long-term solution to addressing the rise in suicides and increased substance abuse among attorneys.

## Law firms

As seen by the results showing heightened alcohol abuse in larger law firms, practice environment is also a factor in this behavior. These large firms themselves must examine their own cultures.

Some of this pressure may come from the production expectations placed on attorneys in these environments. For many large firms, the practice environment, particularly for young attorneys, is defined by billable hours. At most firms, a two thousand (2,000) hour annual billing requirement is considered a minimum. To be deemed successful and receive performance bonuses, attorneys at the largest firms may be expected to bill upwards of two-thousand five hundred hours (2,500) per year [24]. Considering that lawyers on average may only realize as "billable" about a fraction of actual worked hours, this means that for younger, large firm associates, they may potentially "live" their jobs.

While larger firms may need to have a conversation about these expectations and the significance of such expectations on their long-term culture and work life quality, it is doubtful that a fundamental change in this model is possible. Many larger firms depend on this sort of associate production as an integral part of their economic models. Furthermore, this study does not directly link increased alcohol use with younger large firm lawyers much less show a causal link between production expectations and wellness problems.

Particularly with regard to young attorneys, firms must develop a balance between productivity and the wellness of their associates. Psychologists, counselors, and social work practitioners can play an important role in educating firm leadership about the need to develop a sustainable, resilient, and healthy organizational culture. Attorneys themselves must recognize the dangers of a culture that prizes productivity over wellness and be open to reconsideration of longstanding business models.

In fact, some have already suggested that non-traditional practice models, such as part-time schedules or work-from-home options may help attract and keep talented practitioners and may, in fact, yield better work [25]. This is certainly an area warranting further study from both researchers and the firms themselves. Psychologists and counselors can assist in advising on how these models can provide good client services and economic viability while balancing work-life health.

## The bar association

The Bar itself recognizes the need to address the problems. Since receiving the survey results, the Association has redoubled its efforts to increase attorney awareness of the resources available to address quality of life issues. The Bar has increased the frequency of its KYLAP emails to its members. These communications are designed to increase awareness of the resources offered by the Bar to its members. Further, the Bar has devoted greater financial resources to the KYLAP program.

In addition, the Bar has continued to host a virtual town hall series of meetings. Organized by judicial district, these regional meetings are designed to keep members in touch with Bar leadership while fostering a sense of community. During the pandemic these meetings moved virtual, allowing greater attendance. The continued emphasis on communication with the Bar Association and between attorneys themselves has already begun to pay dividends. Mental health and the importance of psychological well-being is a point of emphasis for many of the Bar's subgroups.

For example, the Bar's Young Lawyers Division launched its own program, "Bar the Stigma," an initiative designed to educate members and alleviate preconceptions about mental health care. This is a hopeful development given the increased risks faced by young practitioners. Further, raising awareness among younger lawyers now can only make it easier to make the sort of lasting, cultural change that may help mitigate the problems identified in this study. Mental health professionals have an important role to play in effecting this cultural change. Such professionals must also recognize the current culture of the legal profession and the

challenges it poses to accessing needed wellness resources. After the pandemic's onset, the Bar further implemented a Lawyers Advocating Wellness (LAW) initiative. This new project is a series of awareness and education initiatives designed to raise awareness and offer solutions to members. For example, upcoming LAW programming includes:

- *December 2021*: *FREE Holiday Stress On-Demand CLE presentation*

- *January 2022*: *21 Days Challenge-More information to come*! *Be on the lookout for this activity to start the new year off right.*

- *February 2022*: *Heart Health Month-The KBA will help entice you to get your heart pumping by setting a challenge this month to get our members heart healthy.*

- *March 2022*: *March Nutrition Madness. Ready, set, play*! *What a fun time a year to incorporate a nutrition challenge, as the collegiate level kicks-off the most anticipated month of the year*! *Check social media often for this exciting event.*

- *April 2022*: *Alcohol & Stress Awareness Month-More information and potential programs from the Kentucky Lawyer Assistance Program. Stay tuned.*

- *May 2022*: *KBA Well-Being Bingo*! [26]

Most importantly, member wellness has been a point of focus for the past two KBA Presidents, Hon. Thomas Kerrick and Hon. J. D. Meyer. As stated by Kerrick [27]:

*We must continue to talk about and expose the mental health/suicide issue in our profession in an attempt to address it, as opposed to keeping it in the dark and refusing to admit it. We must get better at recognizing the signs and symptoms and become open to getting treatment for ourselves and our colleagues.*

The Bar appears to be taking the first steps toward a healthier culture. Psychologists and other health professionals must be ready to provide this support.

## Looking forward

While the problems identified in this project still exist, the survey itself was successful in highlighting the need to address stigma related to help-seeking and to acknowledge the multi-layered impact the pandemic placed on attorney well-being. While the lack of some demographic data components are a limitation, it served the purpose of its design. The survey provided important information about the experience of lawyers during the pandemic that has informed the return to post-pandemic "normal," but also lawyer wellness programs. It further raised awareness among the Bar of the ongoing wellness threats to the profession.

Confidentiality continues to be a double-edged sword for the practice of law in wellness issues. While aid received under the KYLAP program is confidential, this same confidentiality makes it difficult to measure the scope of the problem or to always determine if resources are adequately distributed to address the problem.

Most encouragingly, the Bar Association itself is well-aware of the wellness problems of its members, as evidenced by its support of this survey and its efforts in the KYLAP program and its other initiatives. Post-pandemic Bar leadership continues to make lawyer wellness a point of emphasis for Bar programming. However, the work is far from done. The provision of legal services matters to all of our communities. Working to ensure the health and well-being of our legal service providers is a multifaceted problem that will require continued research, education, collaboration and awareness.

## Supporting information

**S1 File.**
(DOCX)

## Author Contributions

**Conceptualization:** Michael Fore.

**Data curation:** Michael Fore, Erin Stevenson.

**Formal analysis:** Michael Fore, Erin Stevenson.

**Investigation:** Michael Fore.

**Methodology:** Michael Fore, Erin Stevenson.

**Project administration:** Michael Fore.

**Resources:** Erin Stevenson.

**Software:** Erin Stevenson.

**Supervision:** Michael Fore.

**Validation:** Michael Fore, Erin Stevenson.

**Visualization:** Michael Fore, Erin Stevenson.

**Writing – original draft:** Michael Fore, Erin Stevenson.

**Writing – review & editing:** Michael Fore, Erin Stevenson.

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
