## [Decision Letter · Decision Letter 0]

16 Nov 2022

PONE-D-22-20142The Impact of Covid-19 Pandemic on Overall Well-being of Practicing LawyersPLOS ONE

Dear Dr. Fore,

Thank you for submitting your manuscript to PLOS ONE. After careful consideration, we feel that it has merit but does not fully meet PLOS ONE’s publication criteria as it currently stands. Therefore, we invite you to submit a revised version of the manuscript that addresses the points raised during the review process.

We look forward to receiving your revised manuscript.

Kind regards,

Pracheth Raghuveer, MD, DNB

Academic Editor

PLOS ONE

https://journals.plos.org/plosone/s/fileid=ba62/PLOSOne_formatting_sample_title_authors_affiliations.pdf.

Reviewers' comments:

Reviewer's Responses to Questions

**Comments to the Author**

1. Is the manuscript technically sound, and do the data support the conclusions?

Reviewer #1: Yes

Reviewer #2: Yes

2. Has the statistical analysis been performed appropriately and rigorously? 

Reviewer #1: I Don't Know

Reviewer #2: I Don't Know

3. Have the authors made all data underlying the findings in their manuscript fully available?

Reviewer #1: Yes

Reviewer #2: Yes

4. Is the manuscript presented in an intelligible fashion and written in standard English?

Reviewer #1: Yes

Reviewer #2: Yes

5. Review Comments to the Author

Reviewer #1: 1. This is an important paper examining an anecdotally well known, yet little researched subject. It adds value to the legal profession and calls for an introspection in the legal community. The issues and recommendations discussed in the paper are very much relevant to jurisdictions beyond where the study was conducted.

2. It is unclear in the paper how location of practice is a determinant of well being. The authors would do well to elaborate on the chosen measures- size of practice and location- as a marker/ determinant of well-being.

3. An intersectional lens accounting for intersectional variability among participants would have added richer insights. The fact that the survey tool was not designed to capture these variances is a big limitation. Well-being and health, access to resources and attrition rates in the legal profession, are intrinsically tied to social location and structural factors. Although the survey tool did not capture these variabilities, the paper would benefit from the discussion section reflecting on structural factors that impact well-being of lawyers.

4. As the author states, the survey was developed primarily to aid the Bar and court's to cope with the pandemic. The recommendations provided are in consonance with the purpose with which the survey was developed. However, a reflection and recommendations at the structural level would add value and richness to the paper.

Reviewer #2: Summary and overall impression:

Fore and Stevenson’s article investigates the impact of the COVID-19 pandemic on legal practitioners in the North American state of Kentucky. The motivation for their paper is to collect evidence of the shared experience of the pandemic. Their evidence consists primarily of quantitative data gathered through an anonymous online survey of 2311 Kentucky bar members. The survey appears to have included a range of questions around mental health and well-being. Overall, the paper empirically highlights the legal profession's mental health crisis. It also exposes the associated stigma, which exacerbates the community's ill-health.

Significance of the study:

The authors have gathered a substantial data set involving 8% of the Kentucky bar, covering various types of practitioners, age groups, and geographical locations. The sample is diverse enough to yield useful insights. As a result, this paper makes an important contribution in the form of empirical data to highlight the obvious but often overlooked problem of mental ill-health in the legal profession.

Major issues:

1) While the study’s relevance is clear, the main argument feels concealed. The authors should expand and clarify the main argument to ensure readers comprehend their research.

2) The paper is a bit unstructured. I would advise the authors to add an outline of the paper in the introduction section. In the conclusion section, I would suggest adding a reiteration of the key findings and an explanation of whether the study was able to fulfil its purpose.

3) The authors write in the discussion section, "increased difficulty with practise was noted by more than three-fourths of all respondents, though public defenders/DPA attorneys appear to have had the most difficulty." It would be helpful if the authors stated whether the PD/DPA reference is in proportion or absolute numbers. Especially since the sample is skewed toward PD/DPAs.

Minor issues:

1) The legal profession is described as "unique" by authors. It would be useful to explain why it is a unique profession and whether there are any characteristics that made the profession especially vulnerable during the pandemic.

2) The authors state in the first paragraph of the Methods section that they were motivated to understand the "shared experience" of legal practise during the pandemic. It would be insightful if the authors investigated the capacity of the shared quality of the experience in addressing stigma associated with mental health.

6. PLOS authors have the option to publish the peer review history of their article (what does this mean?). If published, this will include your full peer review and any attached files.

Reviewer #1: No

Reviewer #2: No

---

## [Author Response · Author response to Decision Letter 0]

3 Dec 2022

PONE-D-22-20142 Response to Reviewers

Reviewer #1: 1. This is an important paper examining an anecdotally well known, yet little researched subject. It adds value to the legal profession and calls for an introspection in the legal community. The issues and recommendations discussed in the paper are very much relevant to jurisdictions beyond where the study was conducted.

2. It is unclear in the paper how location of practice is a determinant of well-being. The authors would do well to elaborate on the chosen measures- size of practice and location- as a marker/ determinant of well-being.

Response- Some additional exposition was added to the methodology sections on practice size and location to better explain the use of these measures. 

3. An intersectional lens accounting for intersectional variability among participants would have added richer insights. The fact that the survey tool was not designed to capture these variances is a big limitation. Well-being and health, access to resources and attrition rates in the legal profession, are intrinsically tied to social location and structural factors. Although the survey tool did not capture these variabilities, the paper would benefit from the discussion section reflecting on structural factors that impact well-being of lawyers.

RESPONSE - - A “looking forward,” section was added to put the survey within context and point the way forward for additional work. 

4. As the author states, the survey was developed primarily to aid the Bar and court's to cope with the pandemic. The recommendations provided are in consonance with the purpose with which the survey was developed. However, a reflection and recommendations at the structural level would add value and richness to the paper.

RESPONSE- A “looking forward,” section was added to put the survey within context and point the way forward for additional work. 

Reviewer #2: Summary and overall impression:

Fore and Stevenson’s article investigates the impact of the COVID-19 pandemic on legal practitioners in the North American state of Kentucky. The motivation for their paper is to collect evidence of the shared experience of the pandemic. Their evidence consists primarily of quantitative data gathered through an anonymous online survey of 2311 Kentucky bar members. The survey appears to have included a range of questions around mental health and well-being. Overall, the paper empirically highlights the legal profession's mental health crisis. It also exposes the associated stigma, which exacerbates the community's ill-health.

Significance of the study:

The authors have gathered a substantial data set involving 8% of the Kentucky bar, covering various types of practitioners, age groups, and geographical locations. The sample is diverse enough to yield useful insights. As a result, this paper makes an important contribution in the form of empirical data to highlight the obvious but often overlooked problem of mental ill-health in the legal profession.

Major issues:

1) While the study’s relevance is clear, the main argument feels concealed. The authors should expand and clarify the main argument to ensure readers comprehend their research.

RESPONSE- Additional language was added to both the abstract and the introduction to emphasize the key findings discussed later in greater detail. 

2) The paper is a bit unstructured. I would advise the authors to add an outline of the paper in the introduction section. In the conclusion section, I would suggest adding a reiteration of the key findings and an explanation of whether the study was able to fulfil its purpose.

RESPONSE- An outline was added at the end of the Introduction section to roadmap out the paper’s contents. A new section was added at the end “looking forward,” to talk about the lasting impact of this work.

3) The authors write in the discussion section, "increased difficulty with practise was noted by more than three-fourths of all respondents, though public defenders/DPA attorneys appear to have had the most difficulty." It would be helpful if the authors stated whether the PD/DPA reference is in proportion or absolute numbers. Especially since the sample is skewed toward PD/DPAs.

RESPONSE- A comment was added to resolve the noted ambiguity.

Minor issues:

1) The legal profession is described as "unique" by authors. It would be useful to explain why it is a unique profession and whether there are any characteristics that made the profession especially vulnerable during the pandemic.

RESPONSE- A paragraph was added to the introduction describing the many roles that lawyers are required to play at different times. 

2) The authors state in the first paragraph of the Methods section that they were motivated to understand the "shared experience" of legal practise during the pandemic. It would be insightful if the authors investigated the capacity of the shared quality of the experience in addressing stigma associated with mental health.

RESPONSE- As mentioned above, additional commentary was added to the introduction to better define the role of a lawyer and the, “shared experience.” Additional clarifying phrases and statements were added in the results and discussion to address this concern.

Thank you to the reviewers for their comments. We believe these revisions have made this a stronger, clearer paper.

---

## [Editor Report · Decision Letter 1]

24 Feb 2023

The Impact of Covid-19 Pandemic on Overall Well-being of Practicing Lawyers

PONE-D-22-20142R1

Dear Dr. Michael Fore

We’re pleased to inform you that your manuscript has been judged scientifically suitable for publication and will be formally accepted for publication once it meets all outstanding technical requirements.

Kind regards,

Shadia Hamoud Alshahrani, PhD

Academic Editor

PLOS ONE

---

## [Editor Report · Acceptance letter]

1 Mar 2023

PONE-D-22-20142R1 

The Impact of Covid-19 Pandemic on Overall Well-being of Practicing Lawyers 

Dear Dr. Fore:

I'm pleased to inform you that your manuscript has been deemed suitable for publication in PLOS ONE. Congratulations! Your manuscript is now with our production department. 

Kind regards, 

on behalf of

Dr. Shadia Hamoud Alshahrani 

Academic Editor

PLOS ONE